# *Oroxylum indicum* (L.) Leaf Extract Attenuates β-Amyloid-Induced Neurotoxicity in SH-SY5Y Cells

**DOI:** 10.3390/ijms26072917

**Published:** 2025-03-23

**Authors:** Nut Palachai, Benjaporn Buranrat, Parinya Noisa, Nootchanat Mairuae

**Affiliations:** 1Biomedical Research Unit, Faculty of Medicine, Mahasarakham University, Maha Sarakham 44000, Thailand; nut.p@msu.ac.th (N.P.); buranrat@gmail.com (B.B.); 2School of Biotechnology, Institute of Agricultural Technology, Suranaree University of Technology, Nakhon Ratchasima 30000, Thailand; p.noisa@sut.ac.th

**Keywords:** baicalein, β-amyloid peptide, chrysin, *Oroxylum indicum* (L.) leaf, oroxylin A, neuroprotection, oxidative stress, SH-SY5Y

## Abstract

Alzheimer’s disease (AD) is characterized by the presence of amyloid-beta (Aβ) plaques, which trigger oxidative stress and neuronal cell death. The present study investigated the neuroprotective effects of *Oroxylum indicum* (L.) leaf (OIL) extract against Aβ-induced oxidative stress and cellular damage in SH-SY5Y cells. The cells were treated with OIL extract with and without Aβ25−35, and their viability was investigated. Moreover, the mechanism of action of OIL was assessed by determining caspase-3 levels, reactive oxygen species (ROS) and malondialdehyde (MDA) levels, enzymatic activity of catalase (CAT), superoxide dismutase (SOD), and glutathione peroxidase (GSH-Px), phosphorylation of phosphoinositide 3-kinase (PI3K)/protein kinase B (Akt), extracellular signal-regulated kinase 1 and 2 (ERK1/2), and cAMP-responsive element-binding protein (CREB), and expression of B-cell lymphoma-2 (Bcl-2) proteins. The results indicated that OIL reduced Aβ-induced neurotoxicity in a concentration-dependent manner, improving cell viability, reducing ROS levels and MDA production, increasing antioxidant enzyme activity of CAT, SOD, and GSH-Px, and decreasing caspase-3 expression. In addition, OIL enhanced phosphorylation of Akt, ERK1/2, and CREB and upregulated Bcl-2 protein expression. High-performance liquid chromatography (HPLC) analysis identified oroxylin A, baicalein, and chrysin as the major phenolic constituents of the OIL extract. The findings suggest that the extract holds promise as a therapeutic intervention against Aβ-induced neurotoxicity, offering potential implications for the treatment of AD. Further studies are needed to investigate the activity of OIL in primary neurons or in vivo.

## 1. Introduction

Alzheimer‘s disease (AD) is an irreversible, progressive neurological disorder that progressively destroys memory and cognitive abilities, ultimately impairing the capacity to perform even the simplest tasks [1]. It is a degenerative disease and the most prevalent form of dementia, characterized by structural and chemical disintegration of the brain, leading to a gradual decline in all aspects of cognition and behavior [1]. Currently, over 6.9 million Americans aged 65 and older are afflicted with Alzheimer’s dementia [2]. The figure may increase to 13.8 million by 2060, assuming that no medical advancements can be employed to prevent or cure AD [2]. The two principal histological indicators of AD, namely extracellular amyloid plaques and intracellular neurofibrillary tangles, are located in vulnerable brain regions, such as the hippocampus and cortex [3]. Amyloid plaques consist of amyloid-beta peptides (Aβ), predominantly including 38–43 amino acids [4,5]. Aβ is a pro-inflammatory and very detrimental substance that induces neuroinflammation in the brain tissue [6]. A growing body of scientific and clinical evidence suggests that elevated Aβ levels may correlate with cognitive impairment and contribute to pathological processes leading to cognitive deficits observed in AD [7,8]. While the specific mechanisms underlying the loss of neurons are still unknown, a number of studies indicate that overproduction of Aβ leads to oxidative stress and free radical production, which in turn cause cell death [9]. Therefore, reducing oxidative stress and thereby moderating or blocking the cytotoxic consequences of Aβ may be a potential approach to treating or preventing AD.

Natural substances and plants have long been used to improve memory and cognition. Traditional medicine uses these ingredients as a possible alternative to pharmaceutical medications, which can be expensive and harmful. They improve memory and cognition mostly owing to neuroprotection. Acting as antioxidants, they combat oxidative stress and reduce inflammation, both of which can damage brain cells and impair brain function. Consequently, the identification of plants that lower oxidative stress and counteract Aβ-induced neurotoxicity is a potential avenue for developing novel AD therapies.

*Oroxylum indicum*, a widely distributed member of the Bignoniaceae family of plants, is also referred to as the “broken bones plant”, “Indian trumpet flower”, “Shyonaka”, and “Midnight horror” [10]. Among the tropical regions which are present, it can also be found in China, the Philippines, Thailand, Taiwan, India, Myanmar, Malaysia, Vietnam, Cambodia, Laos, and Nepal [10]. Asian traditional medicine has long employed *Oroxylum indicum* to treat a wide range of conditions [11]. According to previous research studies, almost every portion of the plant possesses therapeutic qualities, such as anti-inflammatory, antitumor, antioxidant, and immune-suppressive activities [11]. Antibacterial and gastroprotective properties have also been mentioned as advantages [10,11,12]. The flavonoids chrysin, oroxylene A, and baicalein are the main active ingredients of *Oroxylum indicum* [10,11]. Triterpenes, alkaloids, ursolic acid, glycosides, tannins, and carboxylic acids are some of the additional secondary metabolites that have been found. It was recently shown that the fruit pod and seed of *Oroxylum indicum* can reduce oxidative stress and neuroinflammation in BV2 microglial cells [13,14]. Neuroprotective effects of *Oroxylum indicum* pod, bark, and root have also been reported [15,16,17]. Unfortunately, there is a paucity of information regarding the neuroprotective properties of OIL against oxidative stress and cellular damage in SH-SY5Y cells that is caused by Aβ25−35. Since different parts of the plant contain distinct phytochemical compounds that may exert varied effects, the purpose of the current research study was to examine the mechanism by which OIL extract could shield SH-SY5Y cells from oxidative stress and cellular damage caused by Aβ25−35 and to discover the pathways responsible for these neuroprotective benefits.

## 2. Results

### 2.1. Impact of Aβ25–35 on the Survival of SH-SY5Y Cells

Various concentrations (0 to 40 µM) of Aβ25–35 were administered to SH-SY5Y cells for 24 h. The cell viability was determined by the 3-(4,5-dimethylthiazol-2-yl)-2,5-diphenyltetrazolium bromide (MTT) assay. As shown in Figure 1A, cell toxicity induced by Aβ25–35 was noted at a concentration of 20 µM. Therefore, 20 µM Aβ25–35 was selected for further experiments.

### 2.2. Effects of OIL on the Viability of SH-SY5Y Cells

To evaluate the impact of OIL on SH-SY5Y cell viability, cells were subjected to different doses of the extract (12.5, 25, 50, and 100 μg/mL). The concentrations varied from low to high doses to examine both non-cytotoxic and cytotoxic effects. As depicted in Figure 1B, the findings indicated that treatment with 12.5 and 25 μg/mL of OIL extract appeared to increase cell viability relative to the control group; however, the changes were not statistically significant. At a concentration of 50 μg/mL, there was no notable impact on cell viability relative to the control group. However, concentrations up to 100 µg/mL were significantly toxic to SH-SY5Y cells. Therefore, the highest non-toxic concentrations of OIL (25 and 50 μg/mL) were used for subsequent studies.

### 2.3. OIL’s Protective Effect Against Aβ25–35-Induced Cytotoxicity in SH-SY5Y Cells

The cytoprotective effect of OIL against Aβ25–35-induced toxicity was evaluated in SH-SY5Y cells treated with 20 µM Aβ25–35, either alone or in combination with 25 and 50 µg/mL of OIL extract for 24 h. As shown in Figure 2A, treatment with Aβ25–35 alone resulted in a marked reduction in cell density and noticeable structural alterations compared to the control group. Additionally, Aβ25–35 significantly decreased cell viability (*p* < 0.01), as depicted in Figure 2B. In contrast, co-treatment with OIL extract at both concentrations (25 and 50 µg/mL) notably improved cell density and viability in a concentration-dependent manner (*p* < 0.01), demonstrating a significant cytoprotective effect of OIL extract.

### 2.4. OIL Prevents Aβ25−35-Induced ROS Production in SH-SY5Y Cells

Generation of ROS is a known mechanism of Aβ25−35 toxicity [9]. The present study investigated the effect of OIL on intracellular ROS production induced by Aβ25–35. Exposure to Aβ25–35 for 24 h significantly increased ROS levels compared with the control group (*p* < 0.01) (Figure 3A). However, treatment with OIL extract at 25 and 50 μg/mL significantly and concentration-dependently reduced Aβ25–35-induced ROS levels (*p* < 0.01).

### 2.5. OIL Reduces Caspase-3 Expression Induced by Aβ25–35

Caspase-3 is a key marker of neuronal apoptosis [18]. To investigate the protective mechanism of OIL, caspase-3 expression was measured. As illustrated in Figure 3B, treatment with Aβ25–35 for 24 h significantly increased caspase-3 expression compared to the control group (*p* < 0.05). However, co-treatment with 50 μg/mL of OIL extract markedly reduced caspase-3 expression relative to Aβ25–35 treatment alone (*p* < 0.01). Notably, the expression level of the internal control, actin, remained unchanged.

### 2.6. Effects of OIL on Oxidative Stress Status

Table 1 presents the effects of OIL on oxidative stress markers, specifically on the MDA levels and the activities of the following key antioxidant enzymes: CAT, SOD, and GSH-Px. The present study demonstrated that SH-SY5Y cells exposed to Aβ25–35 indicated a significant increase in MDA levels (*p* < 0.05 compared with the control group), indicating elevated oxidative stress. Conversely, the activities of SOD and GSH-Px were significantly reduced (*p* < 0.05 compared with those of the control group), while CAT activity slightly increased in these treated cells. In contrast to these observations, SH-SY5Y cells treated with both Aβ25–35 and OIL at a dose of 50 μg/mL exhibited a significant increase in the activity levels of CAT, SOD, and GSH-Px (*p* < 0.05 for SOD and *p* < 0.01 for CAT and GSH-Px compared with the Aβ25–35 group). In addition, these cells indicated a significant reduction in MDA levels (*p* < 0.05 compared with the Aβ25–35 group).

### 2.7. OIL Increases ERK1/2 and Akt Phosphorylation

The ERK/MAPK and Akt signaling pathways are vital for neuronal processes, such as differentiation, plasticity, and survival. The present study hypothesized that OIL may modulate ERK1/2 and Akt signaling. The experimental results demonstrated that treatment with Aβ25−35 for 30 min significantly decreased p-Akt (Figure 4A) and p-ERK1/2 (Figure 4B) levels compared with the control (*p* < 0.01). However, OIL treatment significantly increased phosphorylation of both Akt and ERK1/2 compared with Aβ25−35 treatment alone. Furthermore, internal control actin did not change.

### 2.8. OIL Enhances CREB Phosphorylation

CREB is a transcription factor involved in neuronal survival and acts downstream of Akt and ERK1/2 signaling. The present study investigated whether OIL could enhance CREB phosphorylation in SH-SY5Y cells treated with Aβ25−35. Following 1 h of Aβ25−35 treatment, CREB phosphorylation was significantly reduced compared with that of the control group (*p* < 0.05). However, OIL treatment significantly increased CREB phosphorylation compared with that of the Aβ25−35 and the control groups (*p* < 0.01). Actin, the internal control protein, also did not change (Figure 5A).

### 2.9. OIL Increases Bcl-2 Expression

CREB has been shown to have a positive regulatory effect on the anti-apoptotic gene Bcl-2, according to previous studies. The purpose of the present study was to determine whether OIL could increase Bcl-2 expression. In the present study, it was discovered that the expression of Bcl-2 was decreased by Aβ25–35 compared with that of the control group (*p* < 0.05). Treatment of the cells with OIL extract for 24 h resulted in a considerable increase in Bcl-2 expression compared with that of the Aβ25−35 group (*p* < 0.05 for 25 μg/mL and *p* < 0.01 for 50 μg/mL). Actin, which controls internal processes, also did not change (Figure 5B).

### 2.10. Analysis of Flavonoid Content in OIL Extract by HPLC

The HPLC analysis of the OIL extract revealed distinct chromatographic fingerprints (Figure 6). Specifically, the peaks corresponding to the compounds baicalein, chrysin, and oroxylin A were identified at retention times of 40.62, 46.89, and 47.84 min, respectively.

These identifications were confirmed by comparing them with the standard mixture HPLC chromatogram shown in Figure 7. The optimized and validated HPLC method was used to quantify the contents of baicalein, chrysin, and oroxylin A in the OIL extract, which were estimated to be 8.00, 2.99, and 26.33 µg/g extract, respectively.

## 3. Discussion

In a previous study, we reported the neuroprotective effects of *Oroxylum indicum* pod extract [15]. However, the neuroprotective properties of OIL extract have not yet been explored. Therefore, this study focuses on the leaf extract, which possesses a distinct phytochemical composition. Our findings demonstrate that the neuroprotective effects of OIL extract are mediated through the ERK/Akt/CREB/Bcl-2 pathways. Additionally, it reduces intracellular ROS and MDA production, boosts antioxidant enzyme activities and modulates the caspase-3 pathway. These new analyses provide valuable insights into the neuroprotective potential of different parts of *Oroxylum indicum* and their mechanisms against beta-amyloid-induced toxicity. In this investigation, the undifferentiated SH-SY5Y cells were selected as a widely accepted model for initial neurotoxicity screening due to their high proliferative capacity and susceptibility to toxic insults. Consistent with previous reports [15], it was observed that Aβ25–35 exhibited cytotoxicity towards SH-SY5Y cells in a concentration-dependent manner. Notably, treatment with OIL significantly increased the cell viability of SH-SY5Y cells exposed to Aβ25–35, suggesting that the OIL extract could shield SH-SY5Y cells from Aβ-induced injury. Given the use of undifferentiated SH-SY5Y cells, we acknowledge the concern that the observed increase in cell viability following OIL extract treatment could be due to a direct protective effect against β-amyloid-induced toxicity rather than enhanced proliferation. However, as shown in Figure 1B and Appendix A, OIL extract treatment at any concentration did not significantly increase cell proliferation compared to the control group, particularly at 50 µg/mL. This suggests that OIL extract does not notably promote cell proliferation. Therefore, the observed increase in cell viability following treatment with both β-amyloid and OIL (Figure 2A,B) is more likely attributable to the neuroprotective effects of OIL rather than enhanced proliferation.

Previous reports have indicated that amyloid plaque formation in the brain tissues of patients with AD causes cytotoxicity through ROS production and increased oxidative stress [9]. The key physiological roles of ROS include the regulation of homeostasis, including redox signaling, defense against pathogens, and protein folding [19,20]. The body’s antioxidant defense system, primarily involving enzymes, such as CAT, SOD, GSH-Px, and glutathione reductase, protects against ROS-induced cellular damage [20,21]. However, when ROS levels become excessive, antioxidant defenses are overwhelmed, leading to oxidative stress. This imbalance disrupts cellular functions by damaging DNA, RNA, amino acids, carbohydrates, lipids, and proteins, particularly affecting neuronal survival [22]. We performed additional assays to further investigate the neuroprotective effects of OIL. Specifically, we assessed oxidative stress markers, including ROS and MDA levels, as well as the activity of antioxidant enzymes such as CAT, SOD, and GSH-Px. The findings confirm that the Aβ treatment enhances oxidative stress by decreasing the activity levels of SOD and GSH-Px, while simultaneously increasing MDA and ROS levels, which are indicators of oxidative stress. Nevertheless, the treatment with OIL extract effectively reversed these changes, indicating that it may improve the functioning of scavenger enzymes and decrease the levels of MDA and ROS. The mitigation of neuronal cell injury and the promotion of neural cell survival were both facilitated by the reduction in oxidative stress, as demonstrated by the decrease in MDA and ROS levels. However, our results showed that Aβ treatment led to an increase in CAT activity compared to the control group, although the difference was not statistically significant. Catalase is a key antioxidant enzyme responsible for breaking down hydrogen peroxide (H_2_O_2_) into water and oxygen, thereby protecting cells from oxidative damage. In AD, Aβ peptides accumulate and trigger oxidative stress in neuronal cells by promoting the excessive production of ROS such as superoxide anions (O_2_^−^), hydroxyl radicals (•OH), and H_2_O_2_. Among these, H_2_O_2_ is particularly harmful due to its ability to diffuse across membranes and generate highly reactive hydroxyl radicals via the Fenton reaction, leading to damage of lipids, proteins, and DNA. The observed increase in CAT activity in Aβ-treated SH-SY5Y cells may reflect a protective cellular response aimed at detoxifying excess H_2_O_2_ and mitigating oxidative stress to preserve cell viability.

Aβ-induced neurotoxicity triggers ROS production and activates caspases [18]. Caspase-3 is a crucial executioner caspase involved in the apoptotic pathway [23]. The caspases are activated in the later stages of apoptosis and are responsible for the cleavage of various cellular substrates leading to the morphological and biochemical changes associated with apoptosis, such as DNA fragmentation and membrane blebbing [23]. The present study specifically investigated the influence of OIL on the expression of this effector caspase. The results indicated that Aβ treatment led to a significant increase in caspase-3 levels; however, OIL (at the concentration 50 μg/mL) administration effectively attenuated this increase.

The Akt is a critical signaling cascade that is responsible for the promotion of cell growth and survival. It is highly expressed in the brain and plays a significant role in neuronal survival [24,25]. The Akt pathway is downregulated by familial AD mutations, as demonstrated by numerous studies. Akt activation facilitates CREB phosphorylation, which subsequently relocates to the nucleus to upregulate the levels of the anti-apoptotic protein Bcl-2 [26]. Because Akt is an upstream regulator of p-CREB and phosphorylation happens quickly, the expression levels of p-Akt and p-CREB were measured at 30 min and 1 h, respectively, while Bcl-2 expression was measured 24 h after treatment. Exposing SH-SY5Y cells to Aβ lowered p-Akt levels, similar with studies of intraneuronal Aβ accumulation leading to diminished p-Akt. [27]. The results also indicated that Aβ exposure decreased CREB phosphorylation and Bcl-2 expression. However, OIL treatment increased Akt and CREB phosphorylation and Bcl-2 expression, suggesting that activation of the Akt/CREB/Bcl-2 pathway is critical for the neuroprotective effects of OIL against Aβ-induced neuronal damage.

The ERK1/2 pathway is another crucial signaling component involved in initiating and regulating cellular processes, such as survival and apoptosis [28]. ERK1/2 promotes cell survival by enhancing the activity of anti-apoptotic molecules; its expression is enhanced in its anti-apoptotic activity by ERK1/2 phosphorylation [28]. In the mammalian cell lines, ERK1/2 signaling can block apoptosis at levels upstream, downstream, or unrelated to the change of mitochondrial transmembrane potential and cytochrome c release [28,29,30]. It has also been reported that CREB functions downstream of ERK, as elucidated in previous studies [31,32]. The present study demonstrated that Aβ exposure led to a notable reduction in p-ERK1/2 levels, while OIL treatment resulted in increased ERK1/2 phosphorylation. These findings suggest that activation of the ERK1/2 pathway is also critical for mediating the neuroprotective effects of OIL against Aβ25–35-induced neuronal damage.

By using an HPLC method, the three major flavonoids, oroxylin A, baicalein, and chysin, were identified as the active components of the OIL extract; this is consistent with a previous study demonstrating that oroxylin A, baicalein, chysin, and baicalin were the major flavonoids in OIL [16,33,34]. Several studies have demonstrated that baicalein and oroxylin A possess significant neuroprotective properties in both in vitro and in vivo models [35]. Previous research studies have shown that baicalein effectively protects against neurotoxicity induced by Aβ, glutamate, 6-hydroxydopamine, hydrogen peroxide, 1-methyl-4-phenylpyridinium, 1-methyl-4-phenyl-1,2,3,6-tetrahydropyridine, and methamphetamine in various cell lines and animal models [36,37,38,39,40,41,42]. In addition, baicalein has been observed to significantly improve cognitive deficits caused by chronic cerebral hypoperfusion by reducing the levels of oxidative stress markers [43]. Baicalein also plays a protective role against Alzheimer’s and Parkinson’s diseases by reducing oxidative stress, preventing the aggregation of disease-specific amyloid proteins, inhibiting excitotoxicity, promoting neurogenesis, and exhibiting both anti-apoptotic and anti-inflammatory effects [44]. Similarly, it has been demonstrated that oroxylin A reduces cognitive impairment brought on by the permanent blockage of bilateral common carotid arteries (2VO) by suppressing activated microglia and upregulating the expression of CREB and brain-derived neurotrophic factor (BDNF) [45]. Chrysin, a bioactive herbal molecule, exhibits a wide range of pharmacological effects, including antioxidant, anti-inflammatory, and neuroprotective properties [44]. Increasing evidence has highlighted the significant role of chrysin in various neurological disorders, such as Alzheimer’s and Parkinson’s diseases [46]. Chrysin has been shown to offer neuroprotection through multiple mechanisms of action, including its antioxidant, anti-inflammatory, and anti-apoptotic functions [46]. In addition, chrysin can indirectly reduce oxidative stress within cells by enhancing the expression levels of key antioxidant enzymes, such as SOD, CAT, and GSH-Px [46]. According to the literature review, there are currently no reports on the effects of baicalein, chrysin, and oroxylin A on beta-amyloid-induced neurotoxicity in SH-SY5Y cells. However, the effect of a single compound may not be as potent as the combined action of multiple compounds. One study demonstrated the neuroprotective effects of *Oroxylum indicum* extract (a combination of baicalein, chrysin, and oroxylin A) on SH-SY5Y cells by upregulating BDNF gene expression under LPS-induced inflammation [16]. Given that HPLC analysis confirmed that baicalein, oroxylin A, and chrysin were the major compounds of OIL, the presence of these compounds in the OIL extract may contribute to the neuroprotection against Aβ-induced neurotoxicity noted in the present study. Nevertheless, additional research might be necessary to compare the effects of each chemical separately and their combined activity from OIL extract on beta-amyloid-induced neurotoxicity in SH-SY5Y cells.

Our study has limited by the fact that it was conducted in vitro and employs cell lines, which may not completely replicate the complexity of primary neurons or in vivo brain environments. Nevertheless, SH-SY5Y cells continue to be a valuable tool for testing compounds for their effects on the nervous system and understanding neurobiology, despite these limitations. Consequently, additional research is required to examine the activity of OIL in primary neurons or in an in vivo model.

## 4. Materials and Methods

### 4.1. Chemicals, Reagents, and Antibodies

Cell culture reagents, including penicillin/streptomycin, were purchased from Hyclone (Logan, UT, USA). The ROS detection kit (catalog number: 4091-99-0), Aβ25−35 (catalog number: A4559-1MG), and MTT kit (catalog number: M5655-1G) were purchased from Sigma (St. Louis, MO, USA). Antibodies that target total Akt (catalog number: A17909), p-Akt (catalog number: AP0637), total ERK (catalog number: A16686), p-ERK (catalog number: AP0974), and actin (catalog number: AC004) were purchased from Abclonal Biotech Co., Ltd.; Wuhan, China. p- CREB (catalog number: AF3189), and BCL-2 (catalog number: AF6139) were purchased from Affinity bioscience; Nanjing, China. The secondary anti-rabbit (catalog number: 34160), anti-mouse antibodies (catalog number: 31430), and the chemiluminescence detection kits (catalog number: 34095) were purchased from thermoscientific; Waltham, MA, USA. Sigma (St. Louis, MO, USA) was also used as a supplier of the RIPA buffer (catalog number: R0278-50ML), protease (catalog number: P2714-BTL), and phosphatase inhibitor cocktails (catalog number: P0044). The bicinchoninic acid (BCA) protein assay kit (catalog number: 23227) was purchased from Rockford, IL, USA. In addition, Merck KgaA, Darmstadt, Germany was the supplier of the malondialdehyde (MDA) detection kit (cataloge number: MAK568). Biovision Inc. of Milpitas, CA, USA, was the supplier of the catalase activity kit (cataloge number: K773-100), and Dojindo Molecular Technologies, Inc., Japan was used as a supplier of the SOD activity kit (cataloge number: 50-190-3738). GSH-Px was sourced from Elabscience Biotechnology Inc., Wuhan, China (cataloge number: E-BC-K096).

### 4.2. Plant Material and Extraction

The leaves of *Oroxylum indicum* (L.) were obtained from MahaSarakham Province, Thailand. The species was identified and confirmed by the Applied Thai Traditional Medicine Department, Faculty of Medicine, Mahasarakham University. The specimen number MSUT_7234 was subsequently preserved in the herbarium of the Faculty of Science, Ma-hasarakham University. The ethanolic extract was obtained by dehydrating, measuring and cutting the leaves, followed by soaking them in 95% ethanol for a duration of 7 days at room temperature. The extract was concentrated using a rotary evaporator, filtered and subsequently lyophilized.

### 4.3. Cell Culture

The SH-SY5Y cell line was obtained from the American Type Culture Collection (ATCC; catalog number CRL-2266; Manassas, VA, USA). The SH-SY5Y human neuroblastoma cell was cultured as previously described. In summary, the cells were maintained at 37 °C in a humidified 5% carbon dioxide (CO_2_) atmosphere in Dulbecco’s Modified Eagle Medium (DMEM) supplemented with 10% fetal bovine serum (FBS), 1% penicillin−streptomycin, and 1% non-essential amino acids. For the experiments, the cells were seeded at optimal densities, and the culture medium was replaced with fresh medium containing Aβ, with or without OIL.

### 4.4. Cell Viability Assay

Cytotoxicity of Aβ25−35 and OIL extract was evaluated in vitro using the MTT assay. The instructions provided were followed to cultivate SH-SY5Y cells in 96-well plates at a concentration of 1 × 10^4^ cells per well. For a 24 h period, the cells were subjected to varying concentrations of Aβ25−35 (ranging from 20 to 40 μM) and OIL extract (at concentrations of 12.5, 25, 50, and 100 μg/mL) in a serum-free medium. The selected OIL extract concentrations were based on dose-response studies, covering a range from low to high doses to evaluate both non-cytotoxic and cytotoxic effects. The non-cytotoxic dose was subsequently chosen for further investigation.

In order to evaluate the protective effects of OIL extract against Aβ-induced neurotoxicity, the cells were subjected to Aβ25−35 alone or in combination with non-cytotoxic dose of OIL extract in a serum-free medium for 24 h. The cells were incubated with MTT reagent (0.5 mg/mL) for 1 h at 37 °C in 5% CO_2_ after the medium was removed post-treatment. Following the removal of the medium, the insoluble purple formazan product was dissolved in 100 μL of dimethyl sulfoxide. The data were expressed as a percentage of the control, and the absorbance at 570 nm was measured using a Synergy-4 plate reader (BioTek Instruments, Inc., Fisher Scientific, St. Louis, MO, USA).

### 4.5. Intracellular ROS Assay

The cells were plated in 96-well plates and cultured as described. Following 24 h of treatment, the cells were incubated with 10 μM CM-2′,7′-dichlorodihydrofluorescein diacetate for 30 min at 37 °C in a CO_2_ incubator. The cells were subsequently washed with PBS and treated with Aβ25−35, with or without OIL extract, in a serum-free medium for 24 h. The Synergy HT Multi-Mode microplate reader (BioTek Instruments, Winooski, VT, USA) was employed to measure the fluorescence intensity of dichlorodihydrofluorescein. The excitation wavelength was 488 nm, and the emission wavelength was 520 nm.

### 4.6. Determination of Oxidative Stress Status

To detect changes in oxidative stress indicators, cells were homogenized in a 0.1 M potassium phosphate buffer solution at pH 7.4 to create cell homogenates. To homogenize the material, dilute 10 mg in 50 µL PBS. The protein concentration in cell homogenates was measured using the Bradford test. MDA levels, as well as activity levels of SOD, CAT, and GSH-Px, were measured using commercial kits in accordance with the manufacturer’s instructions.

### 4.7. Western Blotting

SH-SY5Y cells were cultured in 6-well plates at a density of 1 × 10^5^ cells per well. After treatment, proteins were extracted using RIPA buffer and centrifuged at 10,000× *g* at 4 °C for 10 min to collect the supernatant. Protein concentrations were determined using a Bicinchoninic Acid (BCA) kit. Equal amounts of protein (20 µg per sample) were mixed with a loading buffer, denatured at 95 °C for 5 min and subjected to sodium dodecyl-sulfate polyacrylamide gel electrophoresis (SDS-PAGE).

Following electrophoresis, proteins were transferred onto polyvinylidene fluoride (PVDF) membranes and blocked with 5% skimmed milk in Tris-buffered saline with Tween 20 (TBS-T) (0.1%) for 1 h at room temperature. The membranes were incubated overnight at 4 °C with primary antibodies (1:1000 dilution) against caspase-3, Bcl-2, p-Akt, total Akt, p-ERK1/2, total ERK1/2, p-CREB, and β-actin (used as an internal loading control, 1:5000 dilution). After washing with TBS-T, the membranes were incubated with HRP-conjugated secondary antibodies for 1 h at room temperature. Protein bands were visualized using an enhanced chemiluminescence detection kit, and densitometric analysis was performed to determine fold changes relative to the untreated control.

### 4.8. Analysis of Flavonoids Contents in Crude Extracts by HPLC

The quantitative analysis of three flavonoids in the crude extracts was conducted using the HPLC method. A high-performance liquid chromatographic system was employed utilizing a UHPLC Thermo Scientific model Accela, which included a diode array detector (DAD) from Surveyor and a column heater (Waltham, MA, USA). A Pickering C18 column (150 mm × 4.6 mm; 5 μm particle size) from Pickering Laboratories (Mountain View, CA, USA) was employed. Gradient elution was performed using acetonitrile (solvent A), methanol (solvent B), and 0.01% phosphoric acid in water (solvent C), at a constant flow rate of 800 mL/min. The gradient conditions used were as follows: 0 min: A = 15%, B = 10%, and C = 75%; 30 min: A = 19.5%, B = 11.5%, and C = 69%; 35 min: A = 21%, B = 12%, and C = 67%; 40 min: A = 31%, B = 15%, and C = 54%; 60 min: A = 43%, B = 17%, and C = 40%; 60.1 min: A = 15%, B = 10%, and C = 75%; 63 min: A = 15%, B = 10%, and C = 75%; the column temperature was 25 °C with an injection volume of 10 μL; the detection was performed at 280 nm. The peak area of each standard compound was determined.

### 4.9. Statistical Analysis

Data are presented as mean ± standard error of the mean (SEM) from at least three independent, triplicated trials. Statistical significance was assessed using a Bonferroni post hoc test following a one-way analysis of variance (ANOVA) for comparisons among multiple groups. A *p*-value of less than 0.05 was considered statistically significant. All statistical analyses were conducted using Statistical Package for the Social Sciences (SPSS) software (version 21.0, IBM Corp., Armonk, NY, USA).

## 5. Conclusions

The present study highlights the significant neuroprotective potential of OIL extract against Aβ-induced damage in SH-SY5Y cells. The findings illustrate that OIL operates through several mechanisms (Figure 8), including the reduction in intracellular ROS and MDA levels, enhancement in antioxidant enzyme activities, modulation of the caspase-3 pathway, and activation of critical signaling pathways like ERK1/2 and Akt/CREB/Bcl-2. These outcomes underscore OIL’s role as a promising neuroprotective agent with a multifaceted approach to mitigating Aβ-induced neuronal damage, suggesting its potential therapeutic application in neurodegenerative disorders.

## Figures and Tables

**Figure 1 ijms-26-02917-f001:**
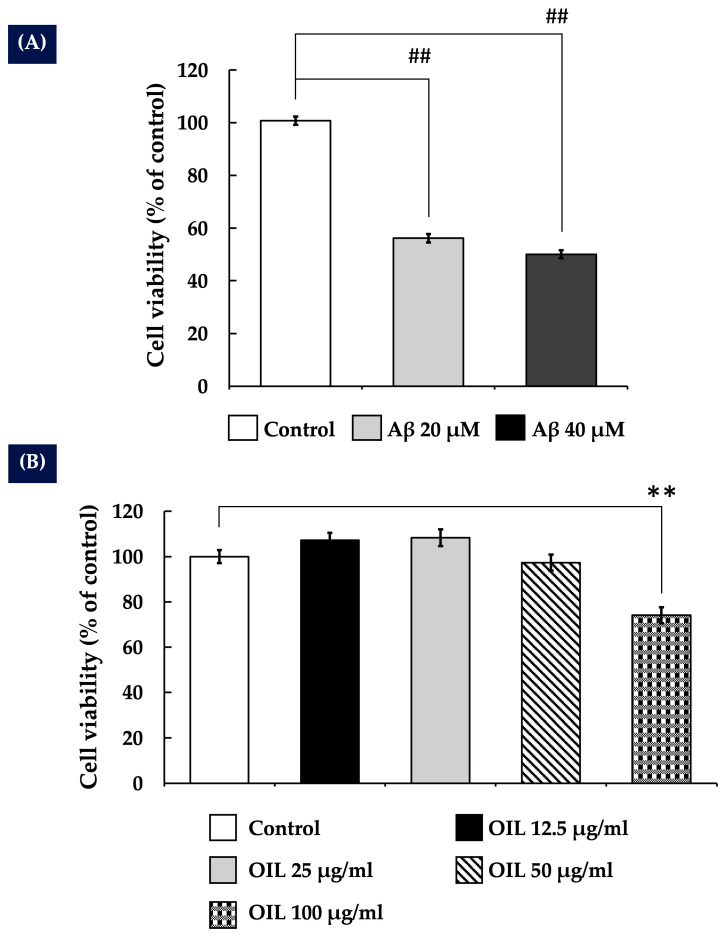
(**A**) Effects of Aβ25−35 on the viability of SH-SY5Y cells. The cells were treated with Aβ25−35 for 24 h, and cell viability was measured by the MTT assay. (**B**) Effects of OIL on the viability of SH-SY5Y cells. The cells were treated with OIL for 24 h, and cell viability was measured with the MTT assay. Data are presented as mean ± SEM of three independent replicates. ** *p* < 0.01 vs. control group; ^##^ *p* < 0.01 vs. control group. Aβ, amyloid-beta peptide; OIL, *Oroxylum indicum* leaf; MTT, (4,5-dimethylthiazol-2-yl)-2,5-diphenyltetrazolium bromide.

**Figure 2 ijms-26-02917-f002:**
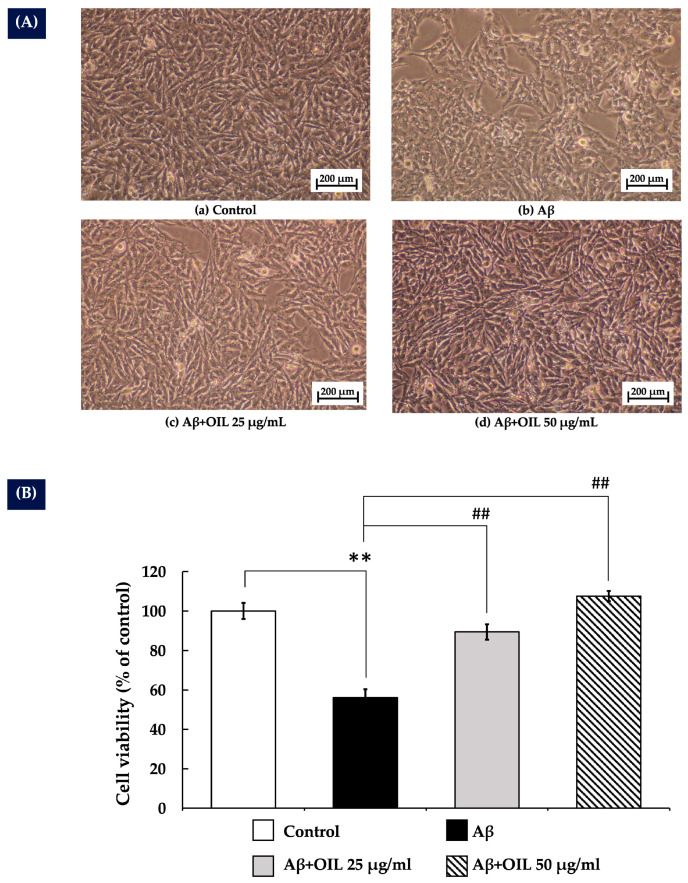
Effect of OIL on Aβ-induced cytotoxicity in SH-SY5Y cells. Treatment was performed for 24 h. (**A**) Light microscopy image of SH-SY5Y cell morphology. (**B**) % cell viability of SH-SY5Y cells determined by the MTT assay. The data are presented as mean ± standard error of the mean from three independent experiments. ** *p* < 0.01 vs. control group; ^##^ *p* < 0.01 vs. Aβ-treated group. Aβ, β-amyloid peptide; Aβ, amyloid-beta peptide; OIL, *Oroxylum indicum* leaf; MTT, (4,5-dimethylthiazol-2-yl)-2,5-diphenyltetrazolium bromide.

**Figure 3 ijms-26-02917-f003:**
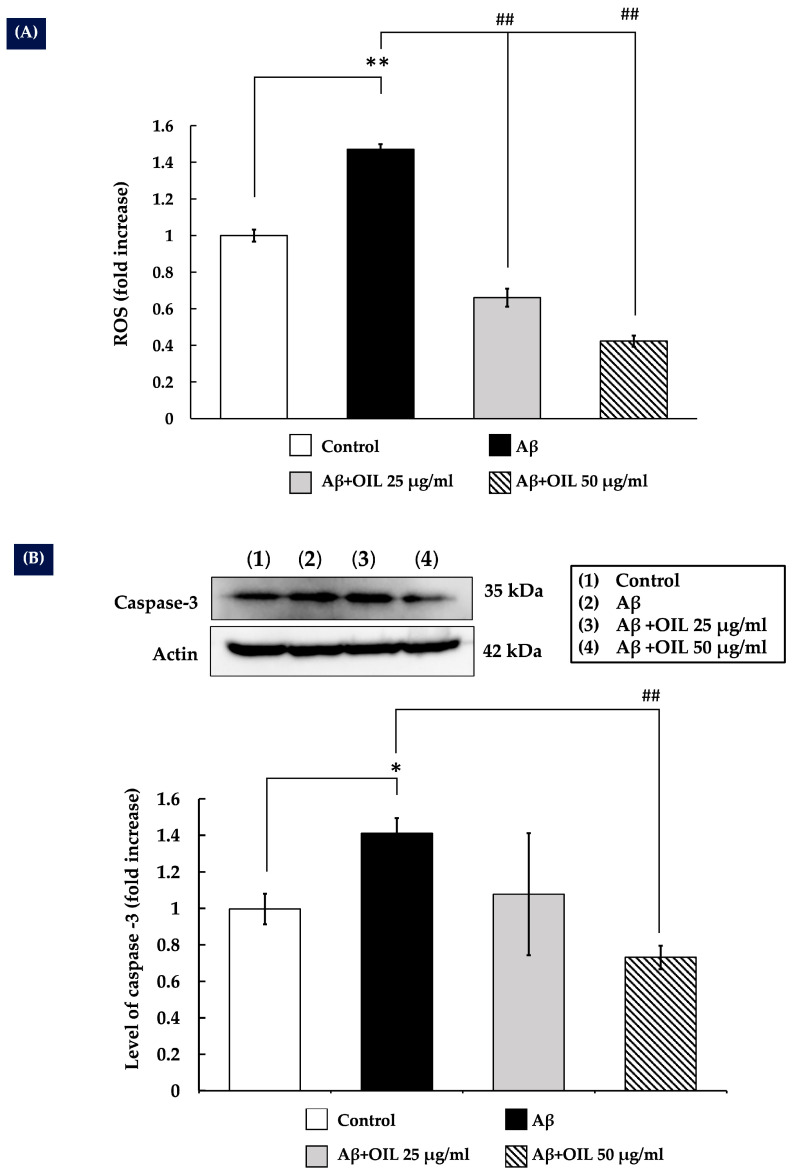
(**A**) Effect of OIL on Aβ-induced free radical production in SH-SY5Y cells, as determined using the fluorescent probe 2’,7’-dichlorofluorescein. (**B**) The impact of OIL on the expression of caspase-3, which is induced by Aβ. Following treatment, total cell lysate was collected, and the levels of caspase-3 expression were determined by Western blotting. Histograms represent the fold-increase relative to that of the untreated control. The data from three separate experiments are shown as mean ± standard error of the mean. * *p* < 0.05 vs. control group; ** *p* < 0.01 vs. control group; ^##^ *p* < 0.01 vs. Aβ-treated group. Aβ, amyloid-beta peptide; OIL, *Oroxylum indicum* leaf.

**Figure 4 ijms-26-02917-f004:**
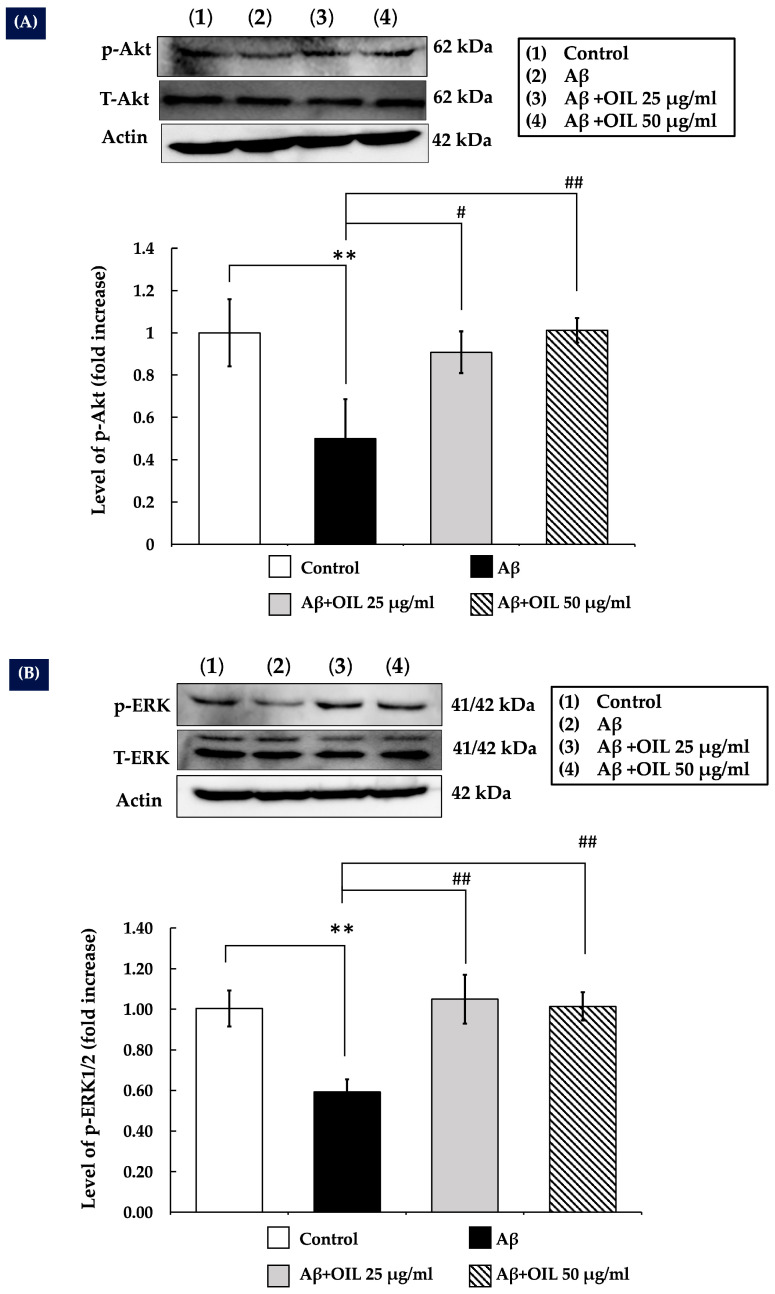
Effects of OIL on p-Akt (**A**) and p-ERK1/2 (**B**) levels. Following treatment, total cell lysate was collected, and the levels of Akt, p-Ak, ERK, and p-ERK were determined by Western blotting. The relative density of p-Akt/Akt and p-ERK/ERK was determined, and the histograms represent the fold increase relative to the untreated control. The data are presented as mean ± standard error of the mean from three independent experiments. ** *p* < 0.01 vs. control group; ^#^ *p* < 0.05 vs. Aβ-treated group; ^##^ *p* < 0.01 vs. Aβ-treated group. Aβ, amyloid-beta peptide; OIL, *Oroxylum indicum* leaf; p, phosphorylated.

**Figure 5 ijms-26-02917-f005:**
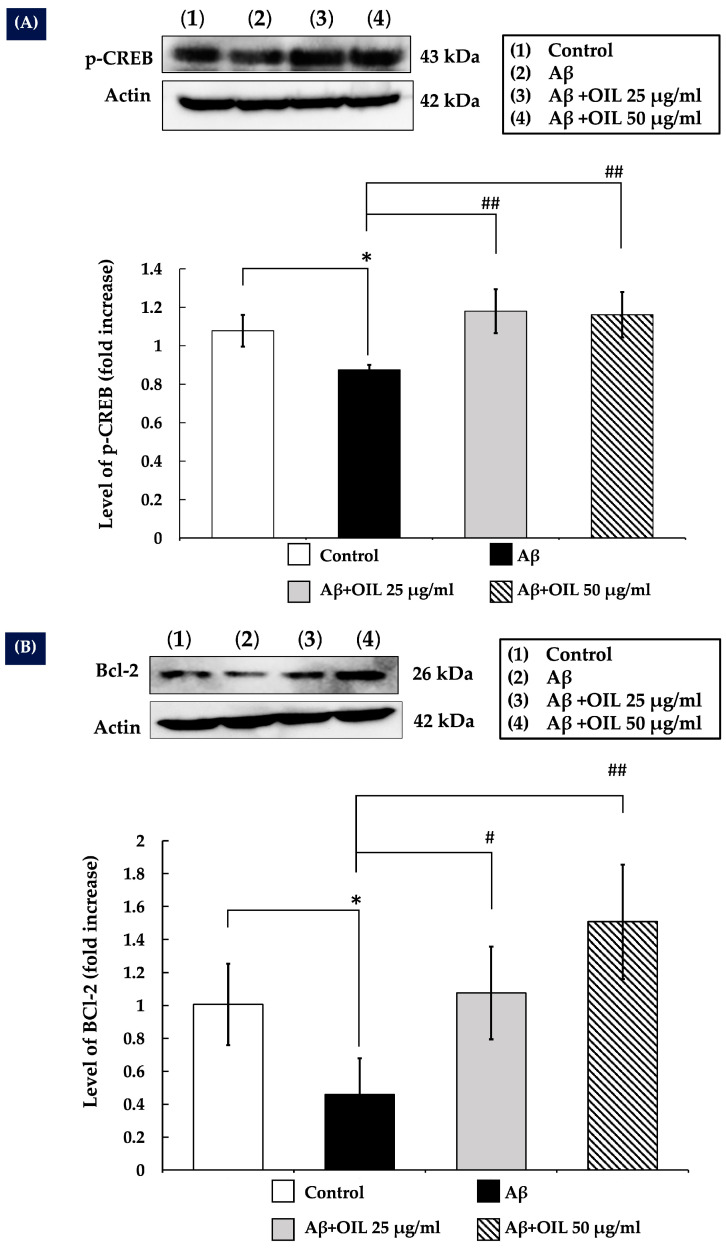
An analysis of the levels of p-CREB (**A**) and Bcl-2 (**B**) expression in response to OIL. The levels of p-CREB and Bcl-2 were evaluated using the Western blot assay after collecting the complete cell lysate, which was performed after treatment. The fold-increase compared to the untreated control is shown in the histograms. The data are represented as mean ± standard error of the mean from three independent experiments: * *p* < 0.05 vs. control group; ^#^ *p* < 0.05 vs. Aβ-treated group; ^##^ *p* < 0.01 vs. Aβ-treated group. Aβ, amyloid-beta peptide; OIL, *Oroxylum indicum* leaf; p, phosphorylated; CREB, cAMP-responsive element-binding protein.

**Figure 6 ijms-26-02917-f006:**
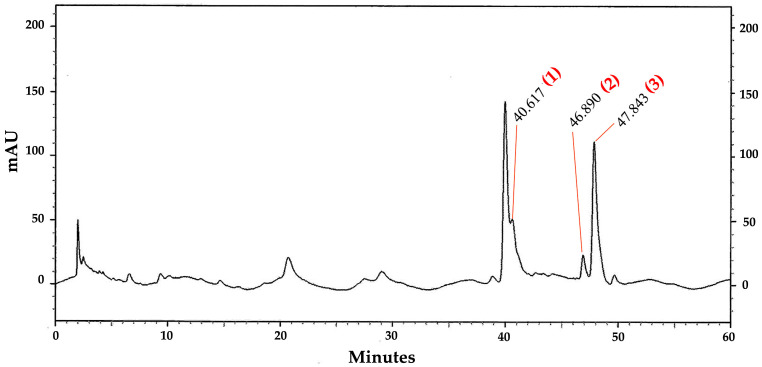
High-performance liquid chromatography chromatograms of OIL extract. Peak 1 indicates baicalein; peak 2 indicates chrysin; peak 3 indicates oroxylin A.

**Figure 7 ijms-26-02917-f007:**
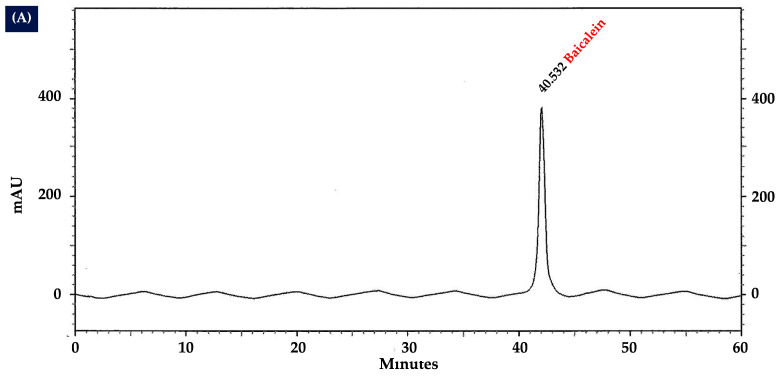
High-performance liquid chromatography chromatograms of standard compounds: (**A**) baicalein; (**B**) chrysin; and (**C**) oroxylin A.

**Figure 8 ijms-26-02917-f008:**
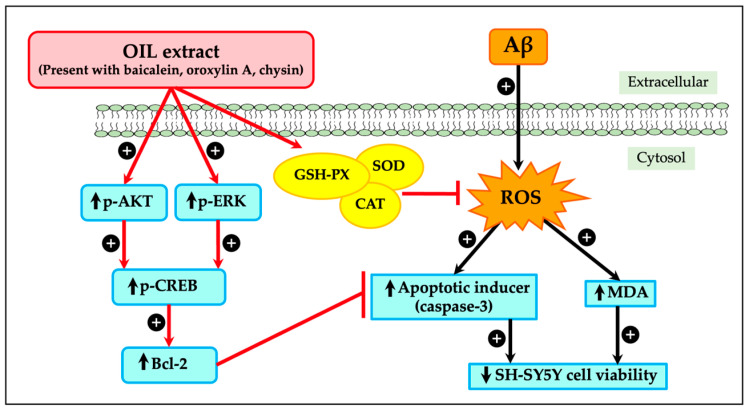
Possible mechanism of OIL-mediated neuroprotection against Aβ-induced cell injury. OIL, *Oroxylum indicum* leaf; Aβ, amyloid-beta peptide; ROS, reactive oxygen species; MDA, malondialdehyde; CAT, catalase; SOD, superoxide dismutase; GSH-Px, glutathione peroxidase; CREB, cAMP-responsive element binding protein.

**Table 1 ijms-26-02917-t001:** The effects of OIL on oxidative stress markers against Aβ25–35-induced cytotoxicity in SH-SY5Y cells.

Treatment Groups	MDA (nmol/mg Protein)	CAT Activity (U/mg Protein)	SOD Activity (% Inhibition/ mg Protein)	GSH-Px Activity (u/mg Protein)
Control	0.569	26.6655	44.670	0.3449
Aβ	0.929 ^a^	30.0036	37.640 ^a^	0.2103 ^a^
Aβ + OIL 25 m/mL	0.882	37.7973 *	44.289	0.4716 *
Aβ + OIL 50 m/mL	0.547 *	60.4452 **	48.139 *	0.6103 **

The data are presented as mean ± SEM. ^a^ *p* < 0.05 compared with the control; * *p* < 0.05 and ** *p* < 0.01 compared with SH-SY5Y cells treated with Aβ. Aβ, amyloid-beta peptides; OIL, *Oroxylum indicum* leaf; MDA, malondialdehyde; CAT, catalase; SOD, superoxide dismutase; GSH-Px, glutathione peroxidase.

## Data Availability

The original contributions presented in this study are included in the article and Appendix A. Further inquiries can be directed to the corresponding authors.

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
