# Peer review of "Oroxylum indicum* (L.) Leaf Extract Attenuates β-Amyloid-Induced Neurotoxicity in SH-SY5Y Cells"

_ijms, 2025, doi:10.3390/ijms26072917_

Round 1
Reviewer 1 Report (New Reviewer)
Comments and Suggestions for Authors Clarity and Structure: The manuscript is well-structured and presents the research findings clearly. However, some sections could benefit from additional detail, particularly the methodology, to enhance reproducibility. Introduction: The introduction provides a good background but could be strengthened by including more recent studies to contextualize the research within current literature. Methodology: The experimental design is sound, but more details on the statistical analysis methods would be beneficial. Consider elaborating on the choice of the specific concentrations of OIL used. Results: The results are presented clearly, with appropriate use of figures and tables. However, Figure 1 could be more detailed to improve interpretability. The histopathology is unclear it need highly magnification and more details. Discussion: The discussion effectively interprets the findings, but it would be helpful to explore potential limitations of the study and suggest future research directions. Conclusion: The conclusion succinctly summarizes the study's contributions. Consider expanding on the practical implications of the findings.Author Response
Response to reviewers and editor suggestion
We sincerely appreciate your letter and the reviewers' valuable comments on our manuscript titled "Oroxylum indicum (L.) Leaf Extract Attenuates β-Amyloid-Induced Neurotoxicity in SH-SY5Y Cells" (Manuscript ID: ijms-3550042).
We are thankful for the opportunity to revise our manuscript and for the insightful feedback provided by the reviewers. Their constructive suggestions have significantly contributed to enhancing the quality and scientific rigor of our work. We have carefully addressed each comment and made the necessary revisions accordingly. Below, we provide a detailed summary of the major modifications along with our responses to the reviewers’ suggestions.
Reviewer 1
Clarity and Structure: The manuscript is well-structured and presents the research findings clearly. However, some sections could benefit from additional detail, particularly the methodology, to enhance reproducibility.
Comments 1: Introduction: The introduction provides a good background but could be strengthened by including more recent studies to contextualize the research within current literature.
Response 1: Thank you very much for your valuable comments and suggestions. The current literature has been updated in the revised manuscript. However, some important older references have been retained.
Comments 2: Methodology: The experimental design is sound, but more details on the statistical analysis methods would be beneficial. Consider elaborating on the choice of the specific concentrations of OIL used.
Response 2: Thank you for your comments and suggestions. We have included additional details on the statistical analysis in the manuscript, highlighted in red on page 17.
Revised Text:
"Data are presented as mean ± standard error of the mean (SEM) from at least three independent, triplicated trials. Statistical significance was assessed using a Bonferroni post hoc test following a one-way analysis of variance (ANOVA) for comparisons among multiple groups. A p-value of less than 0.05 was considered statistically significant. All statistical analyses were conducted using SPSS software (version 21.0, IBM Corp., Armonk, NY, USA)."
Consider elaborating on the choice of the specific concentrations of OIL used.
Response: The concentrations of OIL extract (12.5, 25, 50, and 100 µg/mL) were selected based on dose-response studies. These concentrations span a range from low to high doses, enabling the assessment of both non-cytotoxic and cytotoxic effects. The non-cytotoxic dose was then chosen for further investigation. We have now added a brief justification for this statement in the revised manuscript, highlighted in red on page 15
Revised Text:
“For a 24 h. period, the cells were subjected to varying concentrations of Aβ25-35 (ranging from 20 to 40 μM) and OIL extract (at concentrations of 12.5, 25, 50 and 100 μg/ml) in a serum-free medium. The selected OIL extract concentrations were based on dose-response studies, covering a range from low to high doses to evaluate both non-cytotoxic and cytotoxic effects. The non-cytotoxic dose was subsequently chosen for further investigation.”
Comments 3: Results: The results are presented clearly, with appropriate use of figures and tables. However, Figure 1 could be more detailed to improve interpretability. The histopathology is unclear it need highly magnification and more details.
Response 3: Thank you very much for your comments and suggestions. The results of Figure 1 have been expanded as red fonts on page 3
Revised Text:
“To evaluate the impact of OIL on SH-SY5Y cell viability, cells were subjected to different doses of the extract (12.5, 25, 50, and 100 μg/ml). The concentrations varied from low to high doses to examine both non-cytotoxic and cytotoxic effects. As depicted in Figure 1B, the findings indicated that treatment with 12.5 and 25 μg/ml of OIL extract appeared to enhance cell viability relative to the control group, however the changes were not statistically significant. At a concentration of 50 μg/ml, there was no notable impact on cell viability relative to the control group. However, concentrations up to 100 µg/ml were significantly toxic to SH-SY5Y cells. Therefore, the highest non-toxic concentrations of OIL (25 and 50μg/ml) were used for subsequent studies.”
The histopathology is unclear it need highly magnification and more details.
Response: Thank you for your valuable feedback regarding the histopathology images. In response to your concern, we have adjusted the figure to include higher magnification images to provide more detailed visualization. Specifically, we have enhanced the magnification in Figure 2A and included additional highly magnified images in the supplementary data for further clarity. Moreover, we have expanded the description of the results in the manuscript to provide a more comprehensive explanation, with the revisions highlighted in red on page 4.
Revised Text:
The cytoprotective effect of OIL against Aβ25–35-induced toxicity was evaluated in SH-SY5Y cells treated with 20 µM Aβ25–35, either alone or in combination with 25 and 50 µg/ml of OIL extract for 24 h. As shown in Figure 2A, treatment with Aβ25–35 alone resulted in a marked reduction in cell density and noticeable structural alterations compared to the control group. Additionally, Aβ25–35 significantly decreased cell viability (p < 0.01), as depicted in Figure 3B. In contrast, co-treatment with OIL extract at both concentrations (25 and 50 µg/ml) notably improved cell density and viability in a concentration-dependent manner (p < 0.01), demonstrating a significant cytoprotective effect of OIL extract.
Comments 4: Discussion: The discussion effectively interprets the findings, but it would be helpful to explore potential limitations of the study and suggest future research directions.
Response 4: Thank you for your valuable suggestion. Our limitation was added in this revised manuscript as red fronts, page 14
“Our study has limited by the fact that it was conducted in vitro and employs cell lines, which may not completely replicate the complexity of primary neurons or in vivo brain environments. Nevertheless, SH-SY5Y cells continue to be a valuable tool for testing compounds for their effects on the nervous system and understanding neurobiology, despite these limitations. Consequently, additional research is required to examine the activity of OIL in primary neurons or in an in vivo model.”
Comments 5: Conclusion: The conclusion succinctly summarizes the study's contributions. Consider expanding on the practical implications of the findings.
Response 5: Thank you very much for this comment and suggestion. The conclusion in the revise manuscript indicated as red fronts, page 17
Revised Text:
“The present study highlights the significant neuroprotective potential of OIL extract against Aβ-induced damage in SH-SY5Y cells. The findings illustrate that OIL operates through several mechanisms (Figure 8), including the reduction of intracellular ROS and MDA levels, enhancement of antioxidant enzyme activities, modulation of the caspase-3 pathway, and activation of critical signaling pathways like ERK1/2 and Akt/CREB/Bcl-2. These outcomes underscore OIL's role as a promising neuroprotective agent with a multifaceted approach to mitigating Aβ-induced neuronal damage, suggesting its potential therapeutic application in neurodegenerative disorders.”
Thank you once again for your thoughtful and constructive feedback. We sincerely appreciate the time and effort the reviewers and editor have dedicated to evaluating our manuscript. We have carefully addressed each concern and made the necessary revisions to strengthen the quality and clarity of our work.
Yours sincerely,
Nootchanat Mairuae

Reviewer 2 Report (New Reviewer)
Comments and Suggestions for Authors
The comments are as shown in the attachment.

Author Response
Response to reviewers and editor suggestion
We sincerely appreciate your letter and the reviewers' valuable comments on our manuscript titled "Oroxylum indicum (L.) Leaf Extract Attenuates β-Amyloid-Induced Neurotoxicity in SH-SY5Y Cells" (Manuscript ID: ijms-3550042).
We are thankful for the opportunity to revise our manuscript and for the insightful feedback provided by the reviewers. Their constructive suggestions have significantly contributed to enhancing the quality and scientific rigor of our work. We have carefully addressed each comment and made the necessary revisions accordingly. Below, we provide a detailed summary of the major modifications along with our responses to the reviewers’ suggestions.
Reviewer 2
The article titled “Oroxylum indicum (L.) leaf extract attenuates β-amyloid-induced neurotoxicity in SH-SY5Y cells” presents an investigation into the neuroprotective effects of Oroxylum indicum (L.) leaf extract (OIL) against β-amyloid-induced neurotoxicity using the SH-SY5Y cell line as a model. The authors justify their choice of Oroxylum indicum leaves by noting the limited existing research on this plant part compared to other plant components. The authors have articulated the study's significance, focusing on Alzheimer's disease (AD), a severe neurodegenerative disorder characterized by amyloid-beta (Aβ) plaque formation and oxidative stress and modulation of key signaling pathways such as Akt/ERK/CREB/Bcl-2. The experimental design is logical beginning with preliminary toxicity assays to determine appropriate concentrations for subsequent experiments. The methods employed include standard assays for cell viability (MTT assay), ROS production, caspase-3 expression, oxidative stress markers (MDA, CAT, SOD, GSH-Px), and signaling pathway analyses via western blotting. are suitable for addressing the stated objectives and are commonly used in neuroprotection studies. Overall, this article could contribute to understanding Oroxylum indicum leaf extract's potential neuroprotective effects against Aβ-induced toxicity through multiple mechanisms. The reviewer has the following comments that need to be addressed by authors.
Comments 1. Although the results are generally clear, certain areas require additional clarity or improvement. For instance, while the authors clearly demonstrate that OIL extract reduces ROS production and caspase-3 expression induced by Aβ25–35 treatment, the statistical significance of some intermediate doses (e.g., 25 µg/ml) is not consistently clear across all experiments.
Response 1: Thank you for pointing out this comment. The raw western blot data showed that, at 25 μg/ml of OIL extract, caspase-3 expression was unchanged in two blots (compared to Ab treated group), while one blot showed a decrease. Upon interpretation, the overall result indicated that OIL extract at 25 μg/ml did not significantly reduce caspase-3 expression compared to Ab treated group). Therefore, the sentence “However, while the 25 μg/ml dose showed a decreasing trend, the reduction was not statistically significant” was removed in this revised manuscript.
The remain text at page 5
“As illustrated in Figure 3B, treatment with Aβ25–35 for 24 h. significantly increased caspase-3 expression compared to the control group (p < 0.05). However, co-treatment with 50 μg/ml of OIL extract markedly reduced caspase-3 expression relative to Aβ25–35 treatment alone (p < 0.01). Notably, the expression level of the internal control, actin, remained unchanged.”
Comments 2. While antioxidant enzyme activities were measured comprehensively, the rationale behind unchanged catalase activity in Aβ-treated cells was not adequately discussed or interpreted. The discussion could benefit from deeper exploration into how these specific phytochemicals individually contribute to observed neuroprotection mechanisms.
Response 2: Thank you very much for pointing out this issue. Our results showed that Aβ-treated cells exhibited a slight increase in catalase activity compared to the control group (as demonstrated in table 1). We apologize for the error. It has been corrected and discussed in the revised manuscript, highlighted in red on pages 6.
Revised Text in result:
“Conversely, the activities of SOD and GSH-Px were significantly reduced (P<0.05 compared with those of the control group), while CAT activity slightly increased in these treated cells”.
Text in discussion: page. 12
“However, our results showed that Aβ treatment led to an increase in CAT activity compared to the control group, although the difference was not statistically significant. Catalase is a key antioxidant enzyme responsible for breaking down hydrogen peroxide (H₂O₂) into water and oxygen, thereby protecting cells from oxidative damage. In AD, Aβ peptides accumulate and trigger oxidative stress in neuronal cells by promoting the excessive production of ROS such as superoxide anions (O₂⁻), hydroxyl radicals (•OH), and H₂O₂. Among these, H₂O₂ is particularly harmful due to its ability to diffuse across membranes and generate highly reactive hydroxyl radicals via the Fenton reaction, leading to damage of lipids, proteins, and DNA. The observed increase in CAT activity in Aβ-treated SH-SY5Y cells may reflect a protective cellular response aimed at detoxifying excess H₂O₂ and mitigating oxidative stress to preserve cell viability.”
Comments 3. A notable limitation is the exclusive use of undifferentiated SH-SY5Y cells as a
model system. While these cells are widely used for initial screening due to their proliferative capacity and susceptibility to toxic insults, differentiated SH-SY5Y cells or primary neurons would provide a more physiologically relevant context for assessing neuroprotective effects. Although acknowledged briefly in the abstract's conclusion, this limitation deserves more extensive discussion within the manuscript body.
Response 3: Thank you for your valuable suggestion. Our limitation was added in this revised manuscript as red fronts, page 14 as texts below.
“Our study has limited by the fact that it was conducted in vitro and employs cell lines, which may not completely replicate the complexity of primary neurons or in vivo brain environments. Nevertheless, SH-SY5Y cells continue to be a valuable tool for testing compounds for their effects on the nervous system and understanding neurobiology, despite these limitations. Consequently, additional research is required to examine the activity of OIL in primary neurons or in an in vivo model.”
Comments 4. Isocoumarins that are isomers to oroxylin A, baicalein and chrysin are known for their neuroprotective, antioxidant and anti-inflammatory activities as mentioned in the following articles. The authors are encouraged to cite the given relevant articles. Incorporation of this aspect in the discussion would further strengthen the therapeutic significance of the isolated OIL extract.
https://www.sciencedirect.com/science/article/pii/S0960894X18310047
https://www.sciencedirect.com/science/article/pii/S0223523416307243
Response 4: Thank you very much for this suggestion. It is very interesting papers. However, isocoumarins and flavones are distinct classes of compounds with different core structures. Isocoumarins possess a 1H-2-benzopyran-1-one (1H-isochromen-1-one) structure, while flavones are characterized by a 2-phenylchromen-4-one (2-phenyl-1-benzopyran-4-one) backbone. Oroxylin A, baicalein, and chrysin are flavones, each featuring the 2-phenylchromen-4-one structure with varying hydroxyl groups. In contrast, isocoumarins have a lactonic α-pyranone ring fused to a benzene ring at the 5,6-positions. Therefore, due to these structural differences, isocoumarins are not isomers of flavones like oroxylin A, baicalein, or chrysin. Even if two compounds share the same molecular formula, differences in their structural formulas can result in distinct biological functions. Nevertheless, we have included additional discussion regarding the extraction of oroxylin A, baicalein, and chrysin from OIL for further investigation. (Red fronts, page 13).
However, after reading about isocoumarins, I find them very interesting.
References
- Shabir G, Saeed A, El-Seedi HR. Natural isocoumarins: Structural styles and biological activities, the revelations carry on …. Phytochemistry. 2021;181:112568. doi:10.1016/j.phytochem.2020.112568.
- De Grano RVR, Vashchenko EV, Nisar M, Sung HHY, Vashchenko VV, Williams ID. Crystal structures of the flavonoid Oroxylin A and the regioisomers Negletein and Wogonin. Acta Crystallogr C Struct Chem. 2020;76(Pt 5):490-499. doi:10.1107/S2053229620005550.
- Son SH, Kang J, Ahn M, et al. Synthesis and Biochemical Evaluation of Baicalein Prodrugs. Pharmaceutics. 2021;13(9):1516. Published 2021 Sep 19. doi:10.3390/pharmaceutics13091516.
- Mishra A, Mishra PS, Bandopadhyay R, et al. Neuroprotective Potential of Chrysin: Mechanistic Insights and Therapeutic Potential for Neurological Disorders. Molecules. 2021;26(21):6456. Published 2021 Oct 26. doi:10.3390/molecules26216456.
Comments 5. Certain abbreviations used throughout the manuscript (beyond abstract/introduction) should be defined clearly upon first mention to avoid confusion among readers unfamiliar with specific terminology
Response 5: We greatly appreciate your comment. Abbreviations were referenced throughout the manuscript. New abbreviations were highlight as red fronts.
Thank you once again for your thoughtful and constructive feedback. We sincerely appreciate the time and effort the reviewers and editor have dedicated to evaluating our manuscript. We have carefully addressed each concern and made the necessary revisions to strengthen the quality and clarity of our work.
Yours sincerely,
Nootchanat Mairuae

Reviewer 3 Report (New Reviewer)
Comments and Suggestions for Authors
The manuscript by Nootchanat Mairuae et. al. entitled “Oroxylum indicum (L.) Leaf Extract Attenuates β-Amyloid-Induced Neurotoxicity in SH-SY5Y Cells” provides a comprehensive analysis of the role of OIL in mitigating b-amyloid mediated toxicity. Further, the OIL increases the production of antioxidant enzymes such as CAT, SOD and GSH-Px. Experiments are carefully designed and well-controlled. Including some minor changes mentioned below will further improve the manuscript.
- The authors have shown that the OIL contains baicalein, chrysin and oroxylin A by HPLC (Figure-6). It would be great if authors can describe which compounds out of these three compounds have more prominent effect in SH-SY5Y cells by literature search in mitigating amyloid mediated toxicity in the discussion section.
- In Figure 1 legend, it is not mentioned that data were represented as ± SEM of three independent replicates. Kindly include this line in the Figure-1 legend.
Author Response
Response to reviewers and editor suggestion
We sincerely appreciate your letter and the reviewers' valuable comments on our manuscript titled "Oroxylum indicum (L.) Leaf Extract Attenuates β-Amyloid-Induced Neurotoxicity in SH-SY5Y Cells" (Manuscript ID: ijms-3550042).
We are thankful for the opportunity to revise our manuscript and for the insightful feedback provided by the reviewers. Their constructive suggestions have significantly contributed to enhancing the quality and scientific rigor of our work. We have carefully addressed each comment and made the necessary revisions accordingly. Below, we provide a detailed summary of the major modifications along with our responses to the reviewers’ suggestions.
Reviewer 3
The manuscript by Nootchanat Mairuae et. al. entitled “Oroxylum indicum (L.) Leaf Extract Attenuates β-Amyloid-Induced Neurotoxicity in SH-SY5Y Cells” provides a comprehensive analysis of the role of OIL in mitigating b-amyloid mediated toxicity. Further, the OIL increases the production of antioxidant enzymes such as CAT, SOD and GSH-Px. Experiments are carefully designed and well-controlled. Including some minor changes mentioned below will further improve the manuscript.
Comment 1: The authors have shown that the OIL contains baicalein, chrysin and oroxylin A by HPLC (Figure-6). It would be great if authors can describe which compounds out of these three compounds have more prominent effect in SH-SY5Y cells by literature search in mitigating amyloid mediated toxicity in the discussion section.
Response 1: Your suggestion is greatly appreciated. According to the literature review, there are currently no reports on the effects of baicalein, chrysin, and oroxylin A on beta-amyloid-induced neurotoxicity in SH-SY5Y cells. However, the effect of a single compound may not be as potent as the combined action of multiple compounds. One study demonstrated the neuroprotective effects of Oroxylum indicum extract—a combination of baicalein, chrysin, and oroxylin A—on SH-SY5Y neuronal cells by upregulating BDNF gene expression under LPS-induced inflammation [16]. Given that HPLC analysis confirmed that baicalein, oroxylin A and chrysin were the major compounds of OIL, the presence of these compounds in the OIL extract may contribute to the neuroprotection against Aβ-induced neurotoxicity noted in the present study. However, further investigation is needed to compare the individual effects of each compound and their combined action from OIL extract on beta-amyloid-induced neurotoxicity in SH-SY5Y cells.
In the revised manuscript, this paragraph was included as red front, page 13.
Reference: Sreedharan, S.; Pande, A.; Pande, A.; Majeed, M.; Cisneros-Zevallos, L. The neuroprotective effects of Oroxylum indicum ex- tract in SHSY-5Y neuronal cells by upregulating BDNF gene expression under LPS-induced inflammation. Nutrients 2024, 16(12), 1887.
Comment 2: In Figure 1 legend, it is not mentioned that data were represented as ± SEM of three independent replicates. Kindly include this line in the Figure-1 legend
Response 2: Thank you for your valuable suggestion. In this revised manuscript "Data are presented as mean ± SEM of three independent replicates." was included in figure 1, page 3
Thank you once again for your thoughtful and constructive feedback. We sincerely appreciate the time and effort the reviewers and editor have dedicated to evaluating our manuscript. We have carefully addressed each concern and made the necessary revisions to strengthen the quality and clarity of our work.
Yours sincerely,
Nootchanat Mairuae

This manuscript is a resubmission of an earlier submission. The following is a list of the peer review reports and author responses from that submission.
Round 1
Reviewer 1 Report
Comments and Suggestions for Authors
The presented work explores the neuroprotective effects of Oroxylum indicum (L.) leaf extract against amyloid-beta-induced neurotoxicity in SH-SY5Y cells. The study demonstrates that the extract reduces Aβ-induced oxidative stress and cellular damage by enhancing cell viability, reducing reactive oxygen species and increasing the activity of antioxidant enzymes. Overall, the authors suggest that Oroxylum indicum leaf extract may have potential therapeutic implications for treating Alzheimer's disease.
However, there are some major concerns about the validity of observed findings.
- There is no novelty in the presented manuscript. The experiments are identical to those presented before https://www.spandidos-publications.com/10.3892/mmr.2019.10411
Apart from the 1st point
- Why undifferentiated SH-SY5Y cells were used? They constantly divide, making it hard to conclude if the presented effects are connected to toxicity.
- Connected with previous. Authors claim that OIL increases the vitality of the cells. Since the cells are undifferentiated, it might simply mean they divide faster, not more vital.
- Figure 1 is missing.
- If cells divide faster in the presence of OIL, all observed protective effects might be simply due to a higher number of cells. Proper cell counting should be performed for all presented experiments.
- How was OIL dissolved after lyophilisation?
- What was added as a control to the cells?
Author Response
Response to reviewer and editor suggestion
We are deeply grateful for your letter and the reviewers' insightful comments on our manuscript, "Oroxylum indicum (L.) Leaf Extract Attenuates β-Amyloid-Induced Neurotoxicity in SH-SY5Y Cells" (Manuscript ID: ijms-3498580).
Thank you for the chance to revise our manuscript and for the constructive feedback that was offered. The reviewers' input has been instrumental in improving the quality and scientific rigor of our work, and we genuinely appreciate it. We apologize for any errors in the initial submission.
We have carefully considered each comment and made the necessary revisions to address the concerns raised. Below, we present a detailed summary of the main corrections made and our responses to the reviewers' suggestions.
Response to reviewer 1
The presented work explores the neuroprotective effects of Oroxylum indicum (L.) leaf extract against amyloid-beta-induced neurotoxicity in SH-SY5Y cells. The study demonstrates that the extract reduces Aβ-induced oxidative stress and cellular damage by enhancing cell viability, reducing reactive oxygen species and increasing the activity of antioxidant enzymes. Overall, the authors suggest that Oroxylum indicum leaf extract may have potential therapeutic implications for treating Alzheimer's disease.
However, there are some major concerns about the validity of observed finding.
Comments 1: There is no novelty in the presented manuscript. The experiments are identical to those presented before https://www.spandidos-publications.com/10.3892/mmr.2019.10411
Response 1: Thank you for your comment. While our previous study investigated the neuroprotective effects of Oroxylum indicum fruit extract, the current study focuses on the leaf extract, which has a distinct phytochemical composition. We conducted HPLC analysis to characterize the active compounds in the leaf extract, which was not reported in our previous study on the fruit extract. Additionally, we demonstrated that the neuroprotective effects of Oroxylum indicum leaf extract are mediated through both the AKT and ERK pathways. These new analyses provide valuable insights into the neuroprotective potential of different parts of Oroxylum indicum and their mechanisms against beta-amyloid-induced toxicity.
We have incorporated the additional sentence as indicate in red fronts into the revised manuscript on page 2, lines 76-77, highlighted in red font.
Additionally, we have added the following sentences to the discussion section (highlighted in red font on page 7, lines 203-211):
"In a previous study, we reported the neuroprotective effects of Oroxylum indicum pod extract [15]. However, the neuroprotective properties of Oroxylum indicum leaf extract have not yet been explored. Therefore, this study focuses on the leaf extract, which possesses a distinct phytochemical composition. Our findings demonstrate that the neuroprotective effects of Oroxylum indicum leaf extract are mediated through the ERK/Akt/CREB/Bcl-2 pathways. Additionally, it reduces intracellular ROS and MDA production, boosts antioxidant enzyme activities, and modulates the caspase-3 pathway. These new analyses provide valuable insights into the neuroprotective potential of different parts of Oroxylum indicum and their mechanisms against beta-amyloid-induced toxicity.”
Comments 2: Apart from the 1st point
Why undifferentiated SH-SY5Y cells were used? They constantly divide, making it hard to conclude if the presented effects are connected to toxicity.
Response 2: Thank you for your comment. We selected undifferentiated SH-SY5Y cells as they are a widely accepted model for initial neurotoxicity screening, given their high proliferative capacity and sensitivity to toxic insults. However, we recognize that their continuous division may complicate the interpretation of toxicity-related effects. To address this limitation, we have detected oxidative stress markers, such as ROS and MDA assays, along with caspase-3 analysis, to better assess toxicity in our experiments.
We have incorporated the new sentence into the revised manuscript on page 7, lines 211-213, highlighted in red font.
“In this investigation, the undifferentiated SH-SY5Y cells were selected as a widely accepted model for initial neurotoxicity screening due to their high proliferative capacity and susceptibility to toxic insults.”
Comments 3: Connected with previous. Authors claim that OIL increases the vitality of the cells. Since the cells are undifferentiated, it might simply mean they divide faster, not more vital.
Response 3: Thank you for your insightful comment. Given the use of undifferentiated SH-SY5Y cells, we acknowledge the concern that the increased cell viability observed following OIL extract treatment could be due to a direct protective effect against β-amyloid-induced toxicity rather than enhanced proliferation. However, the MTT assay measures the metabolic activity of viable mitochondria, which correlates with both cell viability and proliferation. To address this concern, we carefully analyzed our data. As shown in Figure 1B, OIL extract treatment at any concentration did not significantly increase MTT absorbance compared to the control group, particularly at 50 µg/mL. This suggests that OIL extract does not notably promote cell proliferation. Therefore, the observed increase in cell viability (color intensity) following treatment with both β-amyloid and OIL (Figures 2A and 2B) is more likely attributable to the neuroprotective effects of OIL rather than enhanced proliferation.
We have added the updated text in the manuscript on page 7-8: lines 217-226, highlighted in red font.
Comments 4: Figure 1 is missing.
Response 4: We acknowledge the reviewer's observation and apologize for this oversight. Figure 1 has been included in the revised manuscript on page 3.
Comments 5: If cells divide faster in the presence of OIL, all observed protective effects might be simply due to a higher number of cells. Proper cell counting should be performed for all presented experiments.
Response 5: Thank you for your comment. In this study, we did not perform cell counting, as explained above. However, we recognize the importance of differentiating between cell proliferation and neuroprotection. To further validate our findings, future studies could include additional experiments, such as cell counting, to provide clearer insights.
Comments 6: How was OIL dissolved after lyophilisation?
Response 6: We dissolved OIL in DMSO, ensuring that the DMSO concentration remained within a non-toxic range for the cells.
Comments 7: What was added as a control to the cells?
Response 7: We added DMSO to the serum-free culture medium as a vehicle control. To ensure that DMSO had no impact on the results, we evaluated its effects by comparing ROS levels and MTT assay results between the control groups treated with serum-free medium alone and serum-free medium with DMSO. No significant differences were observed, indicating that DMSO did not influence the outcome.
Thank you once again for your valuable feedback. We appreciate the time and effort invested by the reviewers and editor in evaluating our manuscript. We have carefully addressed each point raised and made necessary revisions accordingly.
Yours sincerely,
Nootchanat Mairuae

Reviewer 2 Report
Comments and Suggestions for Authors
Palachai and colleagues presented original research entitled " Oroxylum indicum (L.) Leaf Extract Attenuates β-Amyloid-Induced Neurotoxicity in SH-SY5Y Cells"
In this paper, the authors wanted to verify whether a natural substance such as Oroxylym indicum leaf (OIL) can reduce oxidative stress and cellular damage, verifying its neuroprotective action in SH-SY5Y cells treated with Ab25-35.
Overall, the manuscript is somewhat confusing. Some information is missing, even though the data presented suggest that this particular OIL has neuroprotective effects.
In detail:
- Figure 1 has not been included, resulting in some missing information. Panel b is referenced in the legend of Figure 1 but is not discussed in the results.
- MTT is mentioned in the legend of Figure 3; however, the graphs provided do not account for it.
- Provide a more precise explanation of the function of actin in the Western blots (WBs) and clarify whether the actin bands are included in the densitometry shown in the graphs. Additionally, actin is not mentioned in the legend or the materials and methods section.
- Also, in Fig 4, to explain if and how, in addition to actin, the bands of total proteins (tAKT, tERK) are considered. Are they contemplated in the densitometries presented?
- Did the authors verify possible effects on apoptosis in their cellular system, and are these comparable to the data in the literature?
- Furthermore, have you considered verifying the same type of damage and the protective effects of the particular OIL in primary neuronal cultures to overlook the parameters already perturbed in an immortalized line with a tumour origin?
Author Response
Response to reviewer and editor suggestion
We are deeply grateful for your letter and the reviewers' insightful comments on our manuscript, "Oroxylum indicum (L.) Leaf Extract Attenuates β-Amyloid-Induced Neurotoxicity in SH-SY5Y Cells" (Manuscript ID: ijms-3498580 ).
Thank you for the chance to revise our manuscript and for the constructive feedback that was offered. The reviewers' input has been instrumental in improving the quality and scientific rigor of our work, and we genuinely appreciate it. We apologize for any errors in the initial submission.
We have carefully considered each comment and made the necessary revisions to address the concerns raised. Below, we present a detailed summary of the main corrections made and our responses to the reviewers' suggestions.
Response to reviewer 2
Palachai and colleagues presented original research entitled " Oroxylum indicum (L.) Leaf Extract Attenuates β-Amyloid-Induced Neurotoxicity in SH-SY5Y Cells"
In this paper, the authors wanted to verify whether a natural substance such as Oroxylym indicum leaf (OIL) can reduce oxidative stress and cellular damage, verifying its neuroprotective action in SH-SY5Y cells treated with Ab25-35.
Overall, the manuscript is somewhat confusing. Some information is missing, even though the data presented suggest that this particular OIL has neuroprotective effects.
In detail:
Comments 1: Figure 1 has not been included, resulting in some missing information. Panel b is referenced in the legend of Figure 1 but is not discussed in the results.
Response 1: We appreciate the reviewer's observation and sincerely apologize for this oversight.
Figure 1 has been added to the revised manuscript on page 3. Panel 1B is explained in the Results section (Section 2.2) on page 2, lines 86–92.
Comments 2: MTT is mentioned in the legend of Figure 3; however, the graphs provided do not account for it.
Response 2 : We deeply regret the error and are grateful for the reviewer's feedback. MTT (4,5-dimethylthiazol-2-yl)-2,5-diphenyltetrazolium bromide has been removed in the revised manuscript."
Comments 3: Provide a more precise explanation of the function of actin in the Western blots (WBs) and clarify whether the actin bands are included in the densitometry shown in the graphs. Additionally, actin is not mentioned in the legend or the materials and methods section.
Response 3 : Densitometry of actin was measured, and no changes were observed; therefore, we did not include the graph in this study. Since actin is a cytoskeletal protein with stable expression across different conditions and treatments, we used it as a reliable internal control for protein loading and presented only the actin bands.
In the revised manuscript, actin has been included in the methods section (page 9, line 319 and page11, line 389). In the result section, actin was included in page 4, line 213-214; page 5, line 158; page 6, line 181-182.
Comments 4: Also, in Fig 4, to explain if and how, in addition to actin, the bands of total proteins (tAKT, tERK) are considered. Are they contemplated in the densitometries presented?
Response 4 : Thank you for your comment. We have determined the relative density of p-Akt/Akt and p-ERK/ERK, and the histograms illustrate the fold increase relative to the untreated control. These sentences have been included in the revised manuscript on page 5, lines 161–162.
Comments 5: Did the authors verify possible effects on apoptosis in their cellular system, and are these comparable to the data in the literature?
Response 5 : Apoptosis analysis was not conducted in this study. Instead, we specifically measured caspase-3 as an indicator of apoptosis and performed the MTT assay to assess cell viability. These assays allowed us to evaluate the potential neuroprotective effects of OIL extract against β-amyloid-induced toxicity.
Comments 6: Furthermore, have you considered verifying the same type of damage and the protective effects of the particular OIL in primary neuronal cultures to overlook the parameters already perturbed in an immortalized line with a tumour origin?
Response 6 : Thank you for your insightful comment. In this study, we utilized SH-SY5Y cells as a widely accepted in vitro model for neurodegenerative research due to their human origin and neuronal characteristics. The SH-SY5Y human neuroblastoma cell line has been one of the first in vitro models to be developed and has been intensively used to carry on neurotoxicity experiments. However, we acknowledge the limitations of using an immortalized cell line, particularly one with a tumor origin, as it may exhibit altered baseline parameters compared to primary neuronal cultures.
Therefore, we include the sentence “Further studies are needed to investigate the activity of OIL in primary neurons or in vivo.” Page 1, line 27 (in the abstract)
Thank you once again for your valuable feedback. We appreciate the time and effort invested by the reviewers and editor in evaluating our manuscript. We have carefully addressed each point raised and made necessary revisions accordingly.
Yours sincerely,
Nootchanat Mairuae

Round 2
Reviewer 1 Report
Comments and Suggestions for Authors
- In the older version of the manuscript, the authors clearly claimed that “The results revealed that OIL extract increased cell viability at concentrations of 12.5 and 25 μg/ml. “ Now “The results revealed that OIL extract dose not significantly affected cell viability at concentrations of 12.5, 25 and 50μg/ml compared to control group”. It is highly concerning, and since no Figure is provided in the older version, it is impossible to judge which statement is true. Consequently, proper quantification should be present to confirm observed findings. Alternatively, a group with only OIL treatment should be performed for each experiment.
- Also, when presenting the blots on main figures, different repeats are often mixed (e.g. actin from repeat 1 and protein data from repeat 3).
- Why for t-ERK only ERK1 is quantified?
- Why only one band is present for pERK? It should detect 2 bands similar to t-ERK.
- For phospho preteins, it will be better to show the ratio of phospho vs total protein rather than just a change in phospho protein level.
Reviewer 2 Report
Comments and Suggestions for Authors
I thank the authors who considered the suggestions. It can now be accepted for publication.